# Online Simulation Model to Estimate the Total Costs of Tobacco Product Waste in Large U.S. Cities

**DOI:** 10.3390/ijerph17134705

**Published:** 2020-06-30

**Authors:** John E. Schneider, Cara M. Scheibling, N. Andrew Peterson, Paula Stigler Granados, Lawrence Fulton, Thomas E. Novotny

**Affiliations:** 1Avalon Health Economics, Morristown, NJ 07960, USA; john.schneider@avalonecon.com (J.E.S.); cara.scheibling@avalonecon.com (C.M.S.); 2School of Social Work, Rutgers University, New Brunswick, NJ 08901, USA; andrew.peterson@ssw.rutgers.edu; 3School of Health Administration, Texas State University, San Marcos, TX 78666, USA; larry.fulton@txstate.edu; 4School of Public Health, San Diego State University, San Diego, CA 92182, USA; tnovotny@sdsu.edu

**Keywords:** tobacco control policy, tobacco product waste, economic costs of smoking, public policy, environmental policy

## Abstract

Tobacco product waste (TPW) is one of the most ubiquitous forms of litter, accumulating in large amounts on streets, highways, sidewalks, beaches, parks, and other public places, and flowing into storm water drains, waste treatment plants, and solid waste collection facilities. In this paper, we evaluate the direct and indirect costs associated with TPW in the 30 largest U.S. cities. We first developed a conceptual framework for the analysis of direct and indirect costs of TPW abatement. Next, we applied a simulation model to estimate the total costs of TPW in major U.S. cities. This model includes data on city population, smoking prevalence rates, and per capita litter mitigation costs. Total annual TPW-attributable mean costs for large US cities range from US$4.7 million to US$90 million per year. Costs are generally proportional to population size, but there are exceptions in cities that have lower smoking prevalence rates. The annual mean per capita TPW cost for the 30 cities was US$6.46, and the total TPW cost for all 30 cities combined was US$264.5 million per year. These estimates for the TPW-attributable cost are an important data point in understanding the negative economic externalities created by cigarette smoking and resultant TPW cleanup costs. This model provides a useful tool for states, cities, and other jurisdictions with which to evaluate a new economic cost outcome of smoking and to develop new laws and regulations to reduce this burden.

## 1. Introduction

Tobacco product waste (TPW) is one of the most ubiquitous forms of litter, accumulating in substantial quantities on streets, highways, sidewalks, beaches, parks, and other public places, and flowing into storm water drains, waste treatment plants, and solid waste collection facilities [1,2,3,4]. This waste is not simply unattractive; it has been shown to be toxic and costly to clean up [3,5,6,7,8]. In fact, TPW has extensive and unrecognized indirect and direct costs [9], which are addressed in this paper.

More than 249 billion cigarettes were consumed in the United States in 2017 [10]. While many cigarette smokers dispose of their cigarette-related waste properly, it is inevitable that others will simply toss their butts and packages onto streets, beaches, sidewalks, and other public places, thereby creating a public nuisance with negative economic externalities for the costs of cleanup [11]. In an observational study of 9757 individuals in 130 locations in the United States, Keep America Beautiful (KAB) reported a 65% littering rate for cigarette smokers [12]. According to an annual worldwide litter audit performed for more than three decades by the Ocean Conservancy, TPW comprised approximately 37% of all litter by count collected from beaches and coastal areas in 2018 [4]. Some estimates put the total annual weight of TPW in the U.S. at more than 175 million pounds [1,13].

An “externality” occurs whenever the activities of one economic agent affect the activities of another agent in ways that are not taken into account by the operation of the market. When these activities are harmful to one of the economic agents, and the harmed agent is not compensated for the harm, the cause of the harm is typically referred to as a “negative externality” [14,15,16]. The negative economic externalities of smoking on health care costs have been extensively documented [17]. However, the cost of TPW abatement is still relatively poorly understood [7]. Litter is considered a negative externality in that the market prices for litter-producing consumer products generally do not reflect the costs incurred by third parties for the management and disposal of litter—a direct byproduct of consumption of the product. Growing concern over TPW has prompted states and cities to undertake a variety of policy initiatives, including increasing fines and penalties for littering butts, monetary deposits on filters, increasing availability of butt receptacles, assessing fees to cover the public costs of TPW abatement, expanded public education, and implementing extended producer responsibility programs (EPR) [3,7,18,19,20,21,22,23,24,25,26,27,28,29,30,31].

TPW generally originates and accumulates in places where people smoke. As smoking becomes increasingly banned or restricted in restaurants, bars, and public areas such as parks and airports, the locations in which smoking takes place may have become more conducive to discarding cigarette butts [3,32]. When indoor smoking was permitted, ashtrays and other TPW receptacles were typically present in areas where smoking was likely to take place, including dining rooms, bathrooms, lobbies, waiting rooms, and so forth. However, as indoor smoking bans became more prevalent, smoking activities migrated to public areas such as sidewalks, streets, and automobiles (with butts commonly thrown from open windows) [3]. Although many private establishments have attempted to reduce litter in the outdoor areas surrounding their businesses by adding signage and providing TPW receptacles, it is clear from a number of studies that these efforts have had minimal impact on overall TPW accumulation [27,33,34]. Thus, most litter audits conducted in U.S. cities identify the main points of TPW accumulation to be streets, sidewalks, parks, and other open public areas [12,35,36,37]. 

Smoked cigarette butts are the main focus of this analysis, but we note the enormous increase in sales of electronic nicotine delivery systems (ENDS) beginning in 2007 in the United States. These products contain varying amounts of plastics, nicotine, flavorings, and lithium batteries that create additional waste. The tobacco industry and smaller manufacturers have diversified production of these products and have already been able to market some of them with US Food and Drug Administration (FDA) approval, including heat-not-burn products (IQOS). Producers will be submitting for pre-market approval additional tobacco products including ENDS, smokeless tobacco, and various other forms of nicotine and non-nicotine containing products. Currently, we are unable to estimate the total impact such products may have on waste streams, but FDA’s Environmental Assessment on IQOS estimates that modest switching among people who smoke to primarily dual use of the product will result in an increased weight of discarded "sticks" equivalent to 74,100,000 more butts per year (over the previously stated estimate of 249 billion butts per year in 2017) [38]. Further, the liquid nicotine in e-cigarettes is a listed acute hazardous waste product under federal law, and so disposal of ENDS waste may incur additional costs to cities when such waste is collected from public spaces such as schools and government buildings [39]. Finally, the lithium ion batteries in many ENDS create further concern, as these are considered “universal waste,” requiring special handling for household disposal [39,40]. Given the current lack of data on ENDS waste, we have excluded these products from our estimate of TPW costs. Future economic research should include these novel products as they are likely to grow in use throughout the United States. 

To estimate the total TPW cost, we combine (1) indirect costs attributable to the accumulated TPW prior to abatement and (2) the direct costs attributable to deterrence and abatement. In order to fully evaluate the externalities of TPW abatement, we first present a conceptual framework that describes the ways in which TPW generates direct and indirect costs to cities and municipalities. Next, based on this conceptual framework and existing data, we develop an online simulation model to estimate total direct and indirect TPW costs for the 30 largest U.S. cities.

### 1.1. Conceptual Framework 

Estimating a per capita cost for cities could help planning agencies determine how best to manage the costs of TPW and decide whether there are best practices to help cut costs. There have been very few studies performed to estimate TPW costs. However, a 2012 report by Kier Associates for the U.S. Environmental Protection Agency provides a useful model for combining several concepts discussed here to estimate TPW costs [41]. The researchers surveyed a random sample of U.S. West Coast communities in California, Oregon, and Washington located in watersheds that drain to the Pacific Ocean. Communities surveyed ranged in size from very small communities to large cities, such as Los Angeles, California. Respondents were asked to report direct litter mitigation costs in the following categories: beach and waterway cleanup, street sweeping, installation of storm water capture devices, storm drain cleaning and maintenance, manual cleanup of litter, and public anti-littering campaigns [41]. For the largest cities (population ≥ 250,000; *n* = 8), the average annual per-capita costs of litter cleanup were $12.54. The Kier study focuses mainly on waste that may enter a coastal or marine environment, which may be an underestimate of total litter. We increased this estimate in our study by 25%, which resulted in a direct annual cost estimate of $15.68 per person. When adding this amount to the indirect costs that might be associated with TPW, there is an estimated annual cost between $20 and $30 per person.

#### 1.1.1. Indirect Costs-Ecosystems

As a result of smoking behaviors, TPW accumulates in public places. It is picked up either mechanically or manually as part of organized cleanup efforts; what is not picked up remains stationary or migrates into public storm water and sewage systems [42]. Migration to storm and sewer systems can be the result of rain, “hosing down,” “power washing”, or sweeping of sidewalks and streets by municipalities, business owners, or home owners. Sewer accumulation can occur at the point of intake and at numerous screening and filtering points along the storm water and sewer handling and treatment process [43]. It is likely that water-saturated TPW will accumulate in these catchments, whereas unsaturated or less-saturated TPW will float and be carried into the storm and sewer systems and out into streams, rivers, and eventually the aquatic biome. In spite of efforts to mitigate TPW, inevitably a significant proportion of TPW will remain in the environment for some period of time, and its presence has both immediate and cumulative indirect costs.

Nicotine has been used for more than a century as a pesticide and was banned for such use by the United States Environmental Protection Agency in 2014 [44]. It is found in the leachates of TPW and is the most significant component of e-cigarette liquids. Nicotine in TPW and ENDS has been shown to be harmful to marine and aquatic organisms [8].

#### 1.1.2. Indirect Costs—Business and Tourism

In addition to the effects of TPW toxicity to the aquatic biome, environmental “cleanliness” plays an important role in the demand for tourism and as a quality of life issue [1,2,34,45,46,47,48,49]. Urban tourism is to a large extent dependent on “place image”—the process through which individuals perceive and form their impressions of the urban environment. Individuals typically assign high rankings to destination characteristics likely to be affected by TPW, including cleanliness and conservation, trash removal, and beach appearance. Place image also extends beyond tourism; any business can be affected negatively by appearances, including the cleanliness of areas surrounding the entrance to the business. For example, among businesses surveyed in a Florida litter survey, 98% said that the presence of litter lowered property values and had a negative impact on business sales [45].

#### 1.1.3. Direct Costs—Human Health

A growing body of research has shown that cigarette butts have the potential to harm humans and animals [3,6]. Cigarette butts are not “filters” in the technical sense; that is, they do not effectively “filter out” any of the toxins generated in the burning of tobacco and, despite the tobacco industry’s attempt at marketing them as such, they are not a health device and may even cause higher incidence of lung adenocarcinomas [50]. Recent research has found that cigarette butts from smoked cigarettes contain high levels of substances considered toxic to humans and animals, including nicotine, ethylphenol, glycerol, diethylene glycol, propylene glycol, titanium dioxide, glycerol triacetate, other heavy metals, and alkali metal salts of organic acids [5,8,9,51]. The issue of toxicity is particularly relevant because of the migration of TPW to aquatic biomes and public areas such as beaches, bringing greater exposure to children, pets, and other wildlife [6]. There have been documented cases of ingestion of TPW by both children and animals [6]. Additional direct costs of TPW might include emergency room treatments of cigarette butt ingestion by humans and animals, as reported by Novotny et al. [6], as well as costs of damages due to fires caused by discarded butts [52].

#### 1.1.4. Direct Costs—Deterrence

Cities and municipalities devote considerable resources to litter deterrence and abatement. Deterrence includes the posting of signage indicating the fines associated with littering or the harm caused to the environment, and may in some cases include other means through which to increase public awareness, such as public service announcements on billboards and local radio/television broadcasting. Deterrence also includes law enforcement in areas prone to littering and the issuing of summons when law enforcement officials directly observe littering behavior. Littering behavior, however, has still proved to be extremely difficult to address through deterrence alone [12]. Thus, cities must continually engage in litter abatement activities as part of their overall public works responsibilities.

#### 1.1.5. Direct Costs—Abatement

Abatement tasks typically include the following: provision and management of disposal receptacles (general and TPW-specific), mechanical street sweeping, mechanical power washing, manual power washing, manual cleanup, storm drain clean out, and water treatment processes. It should be noted that mechanical street sweeping is not intended solely to collect street litter but is also intended to clear the streets of organic debris (leaves, branches, twigs, sand, gravel, etc.) that would otherwise accumulate on the street and migrate to storm drains. There are also administrative tasks associated with the management and execution of each of these activities. Direct costs generally include costs related to labor (average wages multiplied by total hours worked), equipment and supplies, transportation and fuel, and landfill fees. Administrative costs are also considered direct costs, but such costs are mostly in the form of labor. Mechanical street sweeping and mechanical power washing have relatively high equipment and labor costs (vehicle operators and maintenance), whereas manual power washing and manual cleanup are mainly labor costs. Storm water clean out is mainly labor costs, though specialized equipment may be used in some cases. Water treatment system clean out may in some cases be mechanical (i.e., screens and filters that do not require manual clean out) but may also require a non-trivial labor component. Law enforcement is associated with high labor costs, though the proportion of law enforcement activities attributable to litter deterrence is likely to be very small in most cities and municipalities. Fuel costs can be considerable, especially for the mechanical street sweeping and power washing equipment. Fuel costs are also incurred in transporting litter to landfills, and landfills charge fees based on the weight of the trash deposited.

## 2. Materials and Methods for Simulation Model

### 2.1. Method, Software, Flowchart

The Monte Carlo simulation was used in this study, as exact information regarding some variables (such as per capita litter cleanup costs) is not available. Using known data mixed with probability distributions and sensitivity analysis allows for estimating the total cost ranges for cleanup by city. The simulation was written in R [53] with R Shiny [54] and is available online here: https://rminator.shinyapps.io/TPW1/. The flowchart for the simulation is shown in Figure 1, and steps A through J are explained in the following sections.

### 2.2. Simulation Step A

After a pseudo-random number seed is set for replication, the number of iterations (between 100 and 2000) is set by the user in Step A. This allows for improving the interval estimates based on specific user requirements. Three other parameters are set in Step A, the deterrence/abatement goal (α), the base cleanup cost per person (*B*), and a visitor adjustment factor (*V*).

#### 2.2.1. Deterrence/Abatement Parameter α

In many cases, although cities and municipalities may attempt to clean up 100% of all litter, achievable goals less than 100% are likely more realistic. Cities and municipalities will “tolerate” a certain base amount of litter, such that deterrence and abatement goals are set to an “attainable” percentage level (α). For example, a city may tolerate 10% of existing and accumulated litter, which implies that its goal (α) will be to deter and/or abate 90% of existing and accumulated litter. In step A, the user estimates a global value for α between the range of 0.5 and 1.0.

#### 2.2.2. Per Capita Cost Baseline, B

The third user entry is the deterrence/abatement estimate per capita. The simulation model estimates the total TPW-attributable costs for these 30 largest U.S. cities based on total population residing within city boundaries (Table 1). Baseline litter mitigation costs per capita ($12.54 per capita), are derived from the most comprehensive study (Kier Estimates) of litter mitigation costs to date [41]. However, this survey focuses primarily on litter that can become marine litter, which is an underestimate of total litter. To provide a reasonable range estimate, we increased the Kier estimate by 25% resulting in a baseline direct cost estimate of $15.68 per person. Estimates of the indirect costs of litter are based on the discussion in the preceding section ($6.89 per capita), resulting in a total overall litter cost of $22.56 per capita. Table 1 shows the initial calculations for baseline cost per capita. Given this analysis, we provided the user some flexibility to provide a center estimate of cost per capita between $20 and $30 per person (*B*).

#### 2.2.3. Visitor Adjustment, V

The last user-flexible parameter provides an option to estimate increases or decreases based on visitor nights in the city. Specifically, some large city visitor populations might increase the overall number of daily residents in those cities. Other cities may actually have fewer full-time residents due to seasonal tourism [55]. To account for these differences, the parameter *V* provides between −10% and +10% of a population adjustment.

### 2.3. Simulation Step B, Indices j and k

Step B provides the indices necessary for the simulation. Index *j* is the number of iterations (*j = 1, 2,…N*). Index *k* loops over the cities (*k = 1, 2…30*). An *N × 30* matrix holds the results of the total costs, which are then analyzed post-hoc.

### 2.4. Simulation Step C, Data Sources/Import

The data for the simulation are read in step C. Population data for each city (*POP_k_*) from 2017 derived from the U.S. Census Bureau [56] adult smoking prevalence rates (*PREV_k_*) are from the County Health Rankings data provided by the Centers for Disease Control and Prevention BRFSS dataset [57], and Metropolitan Area Regional Price Parity data for non-rent services (*PARITY_k_*) are from the Bureau of Economic Analysis [58]. All data are publicly available and posted as a table within the simulation.

### 2.5. Simulation Step D, Index for Smoking Prevalence, PREV_k_

City prevalence rates (*PREV_k_*) differ widely, meaning that the proportion of litter waste attributable to TPW varies widely. To account for this factor, an index factor was used in the simulation. The mean prevalence rate for all cities in the study (0.147) calculated by dividing the total number of smokers in all cities (*SMOKERS*) by the population became the denominator to scale each individual prevalence rate as in Equation (1). (NOTE: the denominator accounts for the population differences, which is not much different than the unadjusted mean of 0.153).
(1)$$PREVk=\frac{PREVk}{\sum_{k=1}^{i=30}SMOKERS_{k}/\sum_{k=1}^{i=30}POP_{k}}$$

### 2.6. Simulation Step E, Proportion of Litter that is TPW (λ)

Cities and municipalities do not typically differentiate TPW from general litter in their public works accounting and administration. Thus, estimates of direct costs must be “weighted” to reflect costs associated with TPW as opposed to all general litter. Litter surveys have been conducted in many cities and some cities conduct annual or periodic surveys of litter employing a consistent methodology over time (See Schneider et al. 2011 [7] for a description of the periodic litter surveys conducted by the City of San Francisco) Surveys of “visible” litter can be used to generate an estimate of the percent of all litter that is TPW (for convenience we hereafter refer to this percentage as λ). According to an extensive literature review sponsored by KAB, litter studies conducted around the nation in the past decade have reached remarkably similar findings, with λ estimates ranging from 23% (Toronto) to 37% (Iowa) [1]. The true range of estimates is likely higher; in a national survey of visible litter, KAB found that cigarette butts comprise 36% of all visible litter [35]. In addition, coastal cleanup studies, such as the annual survey conducted by the Ocean Conservancy, generate TPW estimates falling in the range of urban visible litter studies at 37% [59]. Based on these studies, reasonable upper and lower points are available. Given uncertainty around this distribution, λ was modeled as a uniform distribution between 0.2 and 0.4, as this provides a conservative estimate for the cost analysis.

### 2.7. Simulation Step F, Baseline Per Capita Cost Adjustment by City and Uncertainty D_k_

The user-set baseline cost parameter was adjusted for differences in non-rent service costs using Bureau of Economic Analysis (BEA) information for 2017 [58]. BEA data provide information about how much the “average” $100 is worth for these non-rent services. Dividing the original information by 100 provided a reasonable multiplier for deterrence/abatement service costs. Further, lack of knowledge around the user-set baseline parameter provided impetus to model this parameter as a uniform variable between the set baseline minus $1 and plus $1. Equation (2) provides the formula for *D_k_*, the baseline per capita cost adjustment. In this equation, *U* represents the uniform distribution with the minimum and maximum specified within the parentheses.
(2)Dk=U(B−$1, B+$1)×PARITYk

### 2.8. Simulation Step F, Final Cost Per Capita for each City PCC_k_

The final cost per capita, *PCC_k_*, is a straightforward calculation from the previous steps. Equation (3) shows this calculation. The total cost estimate per person from Step E is adjusted for the contribution of TPW waste specified as λ and the municipality goal for deterrence or abatement *α*.
(3)$$PCC_{k}=Dk\times \alpha \times \lambda$$

### 2.9. Simulation Step F, Final Cost for each City TC_k_

The final calculation in the simulation provides the estimate for total cost for each city, *TC_k_*. Total costs are calculated from Equation (3) by multiplying by the population (*POP_k_*) and adjusting for visitor effects on the population (*V*). *V* is designed for evaluating a single city based on its seasonal tourism. Equation (4) is the resultant estimate for city total costs.
(4)TC =Dk×α×λ

### 2.10. Verification, Validation

A combination of methods was used for verification and validation. First, the simulation was initially written in Excel. Those results were evaluated after improvement and migration to R to establish convergent validity. Secondarily, post-hoc results were evaluated for prima facia validity. For verification, the definition of total cost as direct plus indirect cost is well established, and these are the components used.

## 3. Results

### 3.1. Descriptive Statistics, Base Data

Raw data for the simulation are available online at https://rminator.shinyapps.io/TPW1/. Table 2 provides the descriptive statistics for the variables in the model. The median city was of size 877,902, with 123,726 smokers, a prevalence rate of 0.150, and price parity of 0.9865.

A pairs plot (scatterplot matrix) with histograms on the diagonal and bivariate boxplots on the lower triangle shows the distributions and relationships among variables (Figure 2). The upper triangle are correlation plots with correlations stated and loess curves provided to examine linearity. The size of the text indicates the magnitude of the correlation. The correlation of 1.0 between the index and prevalence is based on the fact that index is built from prevalence, smokers, and population variables (see Equation (1)).

For this analysis, we evaluated a simulation with the following parameters: *N* = 2000 simulation, *α* = 0.9 deterrence/abatement goal, *V* = 0% adjustment for visitors, and *B* = $23 base per person cleanup cost. Location statistics and measures of center by city for this simulation are shown in Appendix A. The mean total cost for all cities was $264.5 million, with a median sum of $264.3 million. The mean annual per capita cost was $6.45. Table 3 provides the means and 95% confidence intervals for the total cost estimates by city. The largest interval is less than 0.45% of its associated mean, indicating a narrow range of uncertainty.

### 3.2. Inferential Statistics

Notched boxplots perform a median test, where notches that do not overlap are statistically different at the *p* = 0.05 level (Figure 3). The minimum and maximum estimate seen in 2000 runs for the largest city (New York) was between $38M and $80M, with a median of $58M. For the smallest cities on the list (Portland and Las Vegas), the minimum and maximum were $2M and $4M, with a median near $3M.

## 4. Discussion

To date, there have been no studies specifically designed to assess the indirect costs of TPW. These may include environmental degradation, defined as any change or disturbance to the environment perceived to be deleterious or undesirable. It is also defined by the United Nations as "the reduction of the capacity of the environment to meet social and ecological objectives and needs [60].” In the case of TPW, this waste sullies beaches, neighborhoods, parks and other outdoor recreational areas, thus depriving the population of access to clean, healthful environments. TPW also puts additional disproportionate stress on specific targeted communities by further adding to urban degradation and potentially creating poor perceptions of neighborhoods [61]. Butt flicking is still the social norm for most smokers, and despite evidence showing some reduction in TPW with placement of waste disposal containers, TPW presence on streets and sidewalks suggests that indirect social normative influences against flicking are not yet in play.

The cost pathways for ecosystem impact are less clear, and, as discussed in the conceptual framework, the literature on the association between TPW, ecosystem impact, and human health is relatively new and developmental [6,8,43,51,61,62,63,64,65,66,67,68]. Litter, pollution and industrial accidents and spills can impact the environment in many different ways [69]. The aquatic biome is one of the main pathways through which litter may affect human health. For example, contaminants in the aquatic system impact human health (endocrine, reproductive, genetic, etc.) through ingestion, irrigation, and livestock production. In addition, contaminants can lead to an unbalanced food web and decreased fish and wildlife populations. These effects can, in turn, have an indirect effect on businesses. For example, an unbalanced food web can strain the populations of organisms that organically protect farm crops. Similarly, decreased fish and wildlife populations can negatively impact fishing industries, recreational activities, and tourism [69]. A reasonable assumption might be that the economic impact of the ecosystem effects (including human health) is approximately 50% of the direct effects that are relatively more readily measured (i.e., human ingestion; effect on businesses).

## 5. Conclusions

This study presents the first estimates of the substantial negative economic externalities imposed on large U.S. communities by TPW. In this paper, we have further explored some of the issues that arose in our earlier work on tobacco product waste in San Francisco [7]. In the earlier study, we found that even a relatively narrow scope of litter mitigation activities results in a non-trivial burden on city budgets. The simulation model uses a method considerably different from the methods used in the San Francisco study, but reaches a remarkably consistent estimate. This suggests that our other city estimates are likely to be a reasonable reflection of TPW costs. Cities larger than San Francisco, such as New York, Los Angeles, and Chicago, have more substantial costs—from $27 million in Chicago to nearly $80 million in New York.

This study has some important limitations. First, we use a simulation model to estimate TPW costs, and there are tradeoffs associated with simulation models. The model uses the best available aggregate data and point estimates, but it is not a perfect substitute for a comprehensive survey of TPW costs within each of the U.S. cities. Second, the model is not sensitive to variation in the propensity to litter; the main source of intra-city variation in TPW costs is smoking prevalence, although it is possible that the propensity to litter varies by city or region. Littering is a crime and crime rates vary among cities and regions, and littering rates are likely to vary similarly. Unfortunately, we were not able to identify a reliable and accurate data source on littering propensity. Finally, some cities may be more efficient than others in their cleanup efforts, and this variation in efficiency could result in cost differentials that are not reflected in our model.

As mentioned in the introduction, this model does not include any additional or reductions in TPW costs associated with ENDS and other novel tobacco product use. These may only represent a small percentage of total TPW at present, but it will be important to include in future models the effects of substitution or additional costs associated with these products. Further, in addition to the effects of nicotine mentioned, the lithium in these devices could contribute toxicity to the environment, but this impact is still poorly studied [70].

Nevertheless, economic estimates of the costs of TPW to taxpayers and communities are an important avenue of research that can raise the level of interest in the heretofore poorly managed waste stream of TPW. EPR and PS approaches to TPW costs may have substantial impacts on its prevention, mitigation, and reduction, as these efforts will require tobacco product producers to take responsibility for all or most of the community economic burden due to TPW. EPR/PS approaches may include litter fees and advanced recycling program fees, take back programs such as those for paint and electronics, and litigation for cost recovery [28,29,71]. This may become even more of interest as states, cities, and other jurisdictions increasingly require pre- and post-cost-benefit analyses of new laws and regulations regarding post-consumer waste management of consumer products such as cigarettes and other tobacco products.

## Figures and Tables

**Figure 1 ijerph-17-04705-f001:**
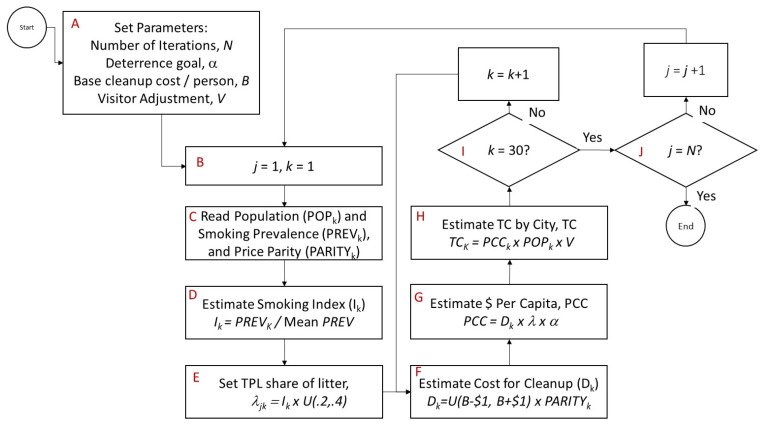
Simulation flowchart for estimating the total cost ranges for tobacco product waste (TPW) cleanup.

**Figure 2 ijerph-17-04705-f002:**
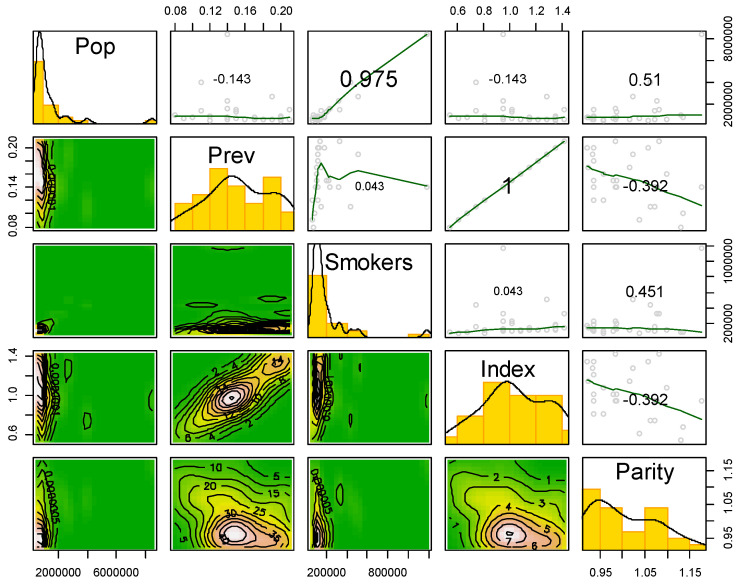
Scatterplot matrix of variables used in the simulation model for estimating the costs of TPW.3.2. Descriptive Statistics, Simulation Run Data.

**Figure 3 ijerph-17-04705-f003:**
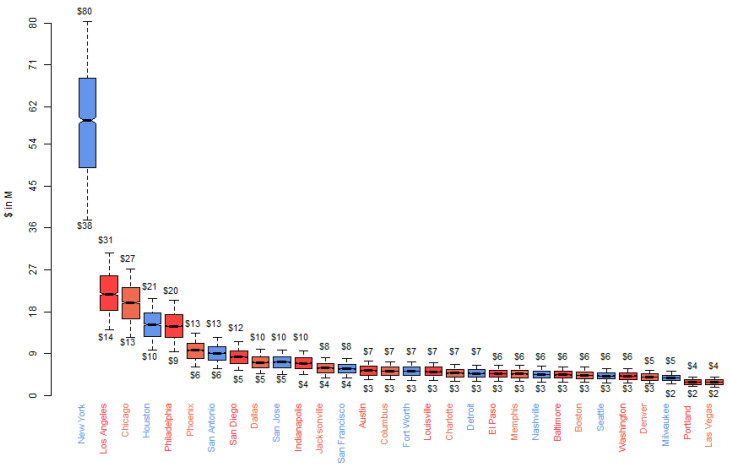
Boxplots of estimated total cost of TPW by City.

**Table 1 ijerph-17-04705-t001:** Baseline Parameter Estimates for TPW Simulation Model in Major U.S. Cities (see previous discussion for sourcing).

Baseline Parameter	Units	Baseline Value
Total Direct (a)	Litter mitigation cost (US$) per capita	$12.54
Total Direct, Adjusted (b)	Litter mitigation cost (US$) per capita	$15.68
Total Indirect (c)	Litter cost (US$) per capita	$6.89
Total Litter Costs (d)	Litter cost (US$) per capita	$22.56
Estimated Baseline TPW Weight (λ) (e)	Percentage weight (0–1)	0.25
Abatement Goal (α) (f)	Percentage weight (0–1)	0.90

(a) Based on large city average from Kier Report to U.S. Environmental Protection Agency (EPA) [41]. (b) Kier Report estimates are based on litter that can become marine litter, which, according to the EPA, is 75% of all litter (thus, adjustment multiplies total direct per capita costs by 1.25); (c) $4.69 per capita (see text) plus an additional 50% associated with ecosystem impact; (d) sum of total direct (adjusted) costs per capita plus total indirect costs per capita; (e) see text on Tobacco Product Waste (TPW) volume calculations; (f) based on the assumption that the typical mitigation goal for cities will be to cleanup 90% of existing litter.

**Table 2 ijerph-17-04705-t002:** Descriptive Statistics of Population, Smokers, Prevalance of Smoking and Price Parity.

Statistic	Pop	Prevalence	Smokers	Parity
Mean	1,364,099.37	0.15	201,454.37	1.01
Standard Error	278,128.30	0.01	38,815.90	0.01
Median	877,901.50	0.15	123,725.50	0.99
Standard Deviation	1,523,371.46	0.04	212,603.44	0.07
Kurtosis	16.39	−0.79	15.69	−0.69
Skewness	3.81	−0.12	3.67	0.63
Minimum	592,025.00	0.08	67,046.00	0.92
Maximum	8,398,748.00	0.21	1,175,825.00	1.18
Sum	40,922,981.00	4.60	6,043,631.00	30.20

**Table 3 ijerph-17-04705-t003:** Means and Confidence Intervals of Total Costs for All Cities based on Simulation Results.

City	Lower CL	Mean	Upper CL
New York	$57,651,833	$58,144,371	$58,636,909
Los Angeles	$19,536,484	$19,703,611	$19,870,738
Chicago	$21,904,786	$22,096,215	$22,287,644
Houston	$14,610,850	$14,736,292	$14,861,734
Philadelphia	$15,031,508	$15,160,748	$15,289,988
Phoenix	$9,497,792	$9,579,344	$9,660,896
San Antonio	$8,983,848	$9,060,898	$9,137,948
San Diego	$7,006,529	$7,066,021	$7,125,513
Dallas	$8,281,272	$8,352,924	$8,424,576
San Jose	$3,875,933	$3,908,981	$3,942,029
Indianapolis	$5,659,452	$5,707,744	$5,756,036
Jacksonville	$7,086,314	$7,146,788	$7,207,262
San Francisco	$4,160,609	$4,195,867	$4,231,125
Austin	$4,518,328	$4,556,811	$4,595,294
Columbus	$6,876,125	$6,935,342	$6,994,559
Fort Worth	$5,140,436	$5,184,420	$5,228,404
Louisville	$4,573,133	$4,611,643	$4,650,153
Charlotte	$4,790,420	$4,831,613	$4,872,806
Detroit	$5,833,292	$5,882,431	$5,931,570
El Paso	$3,738,558	$3,770,661	$3,802,764
Memphis	$5,053,401	$5,097,076	$5,140,751
Nashville	$4,651,138	$4,691,679	$4,732,220
Baltimore	$5,299,099	$5,345,212	$5,391,325
Boston	$4,368,490	$4,405,640	$4,442,790
Seattle	$2,898,986	$2,924,096	$2,949,206
Washington	$5,128,628	$5,172,857	$5,217,086
Denver	$4,407,392	$4,445,049	$4,482,706
Milwaukee	$2,826,926	$2,851,044	$2,875,162
Portland	$4,179,656	$4,215,093	$4,250,530
Las Vegas	$4,654,258	$4,693,621	$4,732,984

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
