# Peer review of "Online Simulation Model to Estimate the Total Costs of Tobacco Product Waste in Large U.S. Cities"

_ijerph, 2020, doi:10.3390/ijerph17134705_

Round 1
Reviewer 1 Report
This article gives us comparability between major US cities for TPW burden, based on specific conceptual framework. Of course there may be some discussion on conceptual framework, especially indirect cost, but under certain assumption, we can see relative TPW burden, between US cities for TPW burden, consist of direct- and indirect- cost from TPW. This is the strength of this article.
Forgive me for pointing a minor points, not related to the main focus of this article.
Among Line 66-73, butts littering and social environmental change (indoor smoking ban) was mentioned, but reference 32 do not seems to be focused on the effect of indoor smoking restriction and butts littering behabiour.
Butts littering is already existing problem before indoor smoking ban.
Reviewer 2 Report
This work evaluated the costs of tobacco product wastes (TPW) in the 30 largest U.S. cities by use of a simulation model. The results are very interesting and helpful for decreasing TPW. However, I would like to conform a point and can’t understand a point. I recommend that authors revise them. The manuscript can be accepted after revision. The points are shown as follows.
1) Table 1, Table 2, Table 3, and Appendix A
How are the numbers of significant figures? The authors should show what digits are accurate.
2) Figure 2.
I have no idea what the figure means. It is necessary to explain in detail.
Reviewer 3 Report
This article seeks to quantify and compare the cost of TPW externalities on thirty American cities, to assist decision making on policies regarding commercial tobacco products and waste management. By putting dollar values on what is often a visible, but ignored, cost to jurisdictions this analysis helps to elevate the importance of TPW management in urban areas.
Generally, there is an issue of terminology that the authors might want to adjust. While the article refers to TPW in its discussion and findings, it seems that the majority of the underlying data regard the most frequently-found source of TPW, cigarette butts. This is understandable, and indeed cigarette butts are the predominate littered TPW, however it does have the potential to mislead readers into thinking this is the "total costs" rather than the costs of the most prevalent form of TPW today. It would be helpful to have some additional discussion within the introduction about how novel products have the potential to be significantly more costly to dispose of correctly. Two specific examples come to mind: e-cigarettes and IQOS.
More specifically on the above point: The liquid nicotine in e-cigarettes is a listed acute hazardous waste under federal law, and so schools, courthouses, and other public entities that accumulate these wastes will be saddled with the compliance costs that come with handling nicotine responsibly. These are costs from TPW that cities will incur, and they will vary significantly based on costs of disposal in each relevant state, but such costs are not incorporated into your modeling. More information on this regulatory burden is available in this government guidance https://environmentalrecords.colorado.gov/HPRMWebDrawerHM/RecordView/434101 and https://rcrapublic.epa.gov/files/14850.pdf. For IQOS, which can now legally be sold in the U.S., FDA's Environmental Assessment on this novel product estimates that modest switching among people who smoke to primarily dual use of the product will result in an increased weight of discarded "sticks" equivalent to 74,100,000 more butts per year. See https://www.fda.gov/media/134458/download (FDA estimated additional solid waste at “a negligible 0.03 percent (by weight) increase in the total litter from all 247 billion cigarette butts consumed in 2017 in the United States.”). That is only one of several heated tobacco products that might change the future of TPW significantly, if FDA continues to approve of such products' sales. Additionally, both e-cigarettes and products like IQOS create a new waste stream of toxic e-waste that was not measured in relevant studies on cigarette butt littering. Although the electronics of e-cigarettes are more likely to be irresponsibly disposed of in the trash (in states like California this is illegal because they are "universal waste" and must be brought to a household hazardous waste facility) instead of littered, this is still a cost to society that indirectly will increase costs on cities.
Additionally, in the "Indirect costs - ecosystems" it might also be worth mentioning that nicotine has been used for more than a century as an effective pesticide, only losing its ability to be sold as such with the EPA in 2014 (https://www3.epa.gov/pesticides/chem_search/reg_actions/reregistration/frn_PC-056702_3-Jun-09.pdf), and that available scientific information shows that e-cigarette liquid is harmful to marine life. See https://pubmed.ncbi.nlm.nih.gov/28957438/ (e-liquid is toxic to Xenopus laevis embryos). Also, the lithium in these devices could be increasing toxic conditions in waterways, but this effect is still poorly studied. https://www.nature.com/articles/s41467-019-13376-y. All lithium-ion batteries studied in one 2013 publication were hazardous waste under California standards. https://pubs.acs.org/doi/abs/10.1021/es400614y.
It also seems that when referencing the "broken window effect" it would be helpful to cite to the Latkin study in endnote 61. The term "broken window" has a negative connotation of being linked to a racist form of community-wide policing, as you allude to. Since that practice has been used to over police the same people who have been targeted the most by the tobacco industry, that reference might be better framed to discuss how TPW puts additional disproportionate stress on specific targeted communities. Consider removing the term.
